# Moderate Water Stress Impact on Yield Components of Greenhouse Tomatoes in Relation to Plant Water Status

**DOI:** 10.3390/plants13010128

**Published:** 2024-01-02

**Authors:** Munia Alomari-Mheidat, Mireia Corell, María José Martín-Palomo, Pedro Castro-Valdecantos, Noemí Medina-Zurita, Laura L. de Sosa, Alfonso Moriana

**Affiliations:** 1Departamento de Agronomía, Universidad de Sevilla, ETSIA, Crta de Utrera Km 1, 41013 Seville, Spain; munia1990@hotmail.com (M.A.-M.); mlpalomo@us.es (M.J.M.-P.); pcvaldecantos@us.es (P.C.-V.); nmedina@us.es (N.M.-Z.); 2Unidad Asociada al CSIC, Uso Sostenible del Suelo y el Agua en la Agricultura (Universidad de Sevilla-IRNAS), Crta de Utrera Km 1, 41013 Sevilla, Spain; 3Instituto de Recursos Naturales y Agrobiología de Sevilla (IRNAS-CSIC) Av. Reina Mercedes 10, 41012 Sevilla, Spain; lauralozano@irnas.csic.es

**Keywords:** deficit irrigation, leaf water potential, stress integral, water stress, water relations

## Abstract

The scarcity of water resources affects tomato production. Deficit irrigation may optimize water management with only a low reduction in yield. Deficit irrigation scheduling based on applied water presented no clear conclusions. Water stress management based on plant water status, such as water potential, could improve the scheduling. The aim of this work was to evaluate the physiological and yield responses of different tomato cultivars to deficit irrigation. Three experiments were carried out in 2020 and 2022 at the University of Seville (Spain). “Cherry” and “chocolate Marmande” cultivars with an indeterminate growth pattern were grown in a greenhouse. Treatments were: Control (full irrigated) and Deficit. Deficit plants were irrigated based on water potential measurements. Moderate water stress did not significantly reduce the yield, although it affected other processes. Fruit size and total soluble solids were the most sensitive parameters to water stress. The latter increased only when persistent water stress was applied. However, truss development and fruit number were not affected by the level of water stress imposed. Such results suggest that moderate water stress, even in sensitive phenological stages such as flowering, would not reduce yield. Deficit irrigation scheduling based on plant water status will allow accurate management of water stress.

## 1. Introduction

Tomato (*Solanum lycopersicum* L.) has been one of the most important vegetables around the world in the last few decades. In fact, the harvested surface increased to more than 5 million ha, a rise of around 30% [1] in the last 20 years. This increment involves much larger irrigation needs, according to published crop coefficients [2]. In addition, cultivation could be limited due to the increasing evaporative demand caused by climatic change. Studies under the conditions experienced in Turkey estimated that a reduction in irrigation water available would strongly lower farmers’ net income, causing even a net loss [3]. Deficit irrigation scheduling would optimize the irrigation water needs under such conditions. However, the yield response to water stress varied depending on the intensity and duration of the deficit period [4]. Similarly, economic return changes were greatly linked to the duration and level of the water stress period but not to the irrigation reduction [3]. While irrigation strategies applying 289.5 mm resulted in a net loss ($-2205 ha^−1^), others providing 151.9 mm generated returns of $942.4 ha^−1^ [3]. These results proved that applied water was not an accurate tool to define deficit irrigation.

Deficit irrigation involves managing water stress at different phenological stages. But water relations in tomato crops are affected by several factors. Tomato species present a great variability of cultivars in terms of fruit size, from small (“cherries”) to very large (“beefsteak”), and plant development (determinate and indeterminate growth). The osmotic adjustment is usually greater in smaller fruits than in bigger ones, and it increases drought tolerance [5]. The phenological stages could be better defined in determinate growth crops than in plants of indeterminate growth, in which flowering, fruit growth, and vegetative growth occur simultaneously. The effect of the root signal could also enhance the variability of the response to deficit conditions. Partial root drying (PRD) has been reported as a very interesting irrigation method in tomatoes [6,7,8,9]. PRD involves changing the conditions of different zones in tomato roots periodically, so some parts can be wet or dry, depending on the scheduling. Such changes produced a root signal, which allowed a better adaptation to water stress with a lower decrease in water potential and less reduction in assimilation [9]. Thus, PRD improved the yield response with the same amount of water as traditional deficit irrigation [6,7,8,9]. However, the PRD response was affected by the frequency of dry/wet zone changes [6] and by the amount of water uptake during the dry period [10]. Therefore, all these responses showed that, in fact, the pattern of water stress was the most important factor in define yield response when using this irrigation method.

Deficit irrigation was commonly applied as a sustained deficit throughout the season. Sustained irrigation works reported that applying around 50% of the full irrigated water did not significantly reduce yield [6,9,11]. However, other experiments suggested that applying anything below 75% would affect yield [12,13]. This lack of results suggests that applied water is not the most accurate tool to manage deficit irrigation if water stress is not measured. A recent meta-analysis study on tomatoes [14] concluded that soil texture was an important factor to consider in the results of irrigation work. Then, the influence of soil water stored and the lack of conclusive results in some sustained irrigation works showed the importance of the time point and the level of water stress imposed. The best deficit irrigation strategy would decrease the amount of water applied during the vegetative period, causing a small or no impact on yield [3,15,16]. Conversely, the period of fruit set and fruit growth were reported to be the most sensitive to water stress in relation to yield [3,11,15,16]. Finally, the ripening period would be an intermediate phenological stage in terms of drought resistance because, although yield could be significantly reduced [3,15,17], total soluble solids would also be enhanced [17], which is a very interesting fruit feature for processing cultivars. However, these latter results would be more precise when the water stress level was considered. The lack of yield reduction due to water stress during the flowering period [18,19], a very sensitive phenological stage, shows the importance of the water stress level in addition to the time point.

The characterization of water stress has not been common in deficit irrigation projects for tomato crops. The soil matrix potential has been suggested as a useful tool in these strategies. Thompson et al. [20] proposed values between −38 and −58 KPa for tomato crops in the spring and autumn cycles, respectively. But Marouelli and Silva [21] changed the recommendation and based it on the phenological stage: −35 KPa in the vegetative stage, −12 KPa in the fruit development stage, and −15 KPa in the ripening stage. The position of sensors in those works was very important as well. In the latter publication, changes in the threshold of the soil matric potential were coupled with an increase in depth from 10 to 20 cm [21]. This limitation could be overcome using plant water status measurements, which would include all factors affecting water relations and yield responses to deficit irrigation [5]. The tomato crop presented a non-isohydric pattern under water stress conditions, with a fast decrease in water potential before the gas exchange reduction [22]. However, the threshold values of leaf water potential and the responses of truss and fruit development were not commonly described. Values of midday leaf water more negative than −1 MPa have been reported to decrease yield [7,9,15,23,24]. However, in some of these latter works, the seasonal pattern of water potential measurements was unknown [15,23]. Moreover, in other works conducted in rainy places [18] or in the autumn cycle [24], this threshold did not affect yield. Also, the cumulative effect of water stress could be more useful for determining deficit irrigation than isolated minimum values. Rudich et al. [25] suggested the “stress factor” as a useful indicator to manage deficit irrigation in tomato crops. The stress factor is the area limited by the measured values of the leaf water potential seasonal pattern and a constant value. Such an approach would include the level and duration of water stress [26]. Changes to the constant value, which is the reference, would affect the result obtained. Rudich et al. [25] used −0.6 MPa as a threshold because they considered this value to be the beginning of severe water stress conditions. Coyago-Cruz et al. [24] suggested using the stress integral, with the most positive value measured in the experiment (−0.2 MPa in this work), as recommended by Myers [26]. In fact, this reference value would be variable because the leaf water potential was related to the evaporative demand [4]. In other fruit crops, this and other plant water status indicators were considered reference equations of full irrigated treatments to decrease the variations due to evaporative demand [27,28,29]. These measurements of the plant water status, in addition to plant development, would enhance knowledge about water stress management in tomatoes.

The description of water status for tomato plants related to deficit irrigation management is not frequent in the literature. This could limit the recommendations resulting from experimental conditions. The cumulative effect of water stress was also reported less frequently in tomato and vegetable production in general. Both issues, in addition to the management of the application time point, could improve deficit irrigation scheduling. Moreover, these effects could change with the growth cycle and the fruit size, which are very variable in tomato cultivars. Therefore, the aim of this work was to evaluate the response of yield and yield components to deficit irrigation scheduling, considering the plant water status. This evaluation considered not only the effect of water stress on total yield but also some quality parameters and the main yield components, such as the number of fruits, fruit size, and truss development. The hypothesis was that using different cycles and fruit sizes would allow a wide range of yield vs. water status responses, which could improve the deficit irrigation scheduling recommendations.

## 2. Results

### 2.1. Water Relations and Vegetative Growth

Figure 1 shows the seasonal pattern of applied water. Data in the spring cycles were increasing in Control plants, mainly from 80 DAT.

The maximum amount of water was similar in both seasons, with values being 613 and 687 mm in 2020 and 2022, respectively. The seasonal pattern of autumn 2022 was linear until 80 DAT, when Control plants were hardly irrigated. In this latter experiment, the applied water during the treatment was 211 mm, clearly lower than in the other two. Conversely, the Deficit treatment presented a different pattern between cycles. During 2020, several periods of no irrigation occurred throughout the season, although several irrigation events took place until 140 DAT. In this experiment, the total amount of applied water reached 290 mm, around 47% of the Control. In Spring 2022, irrigation was reduced in the Deficit treatment, with a short period of irrigation from 80 to 103 DAT. In this season, the total amount of applied water was 88 mm or around 13% of Control plants. Finally, the most restrictive experiment took place in autumn 2022. In this season, only a short irrigation event took place between 53 and 64 DAT (4 mm). The total amount of applied water was 32 mm or around 15% of Control plants.

The pattern of midday leaf water potential (WP) was different in each year and cycle (Figure 2). In the Spring cycles, the most negative WP values were more frequent than in the autumn cycle for both treatments, although the amount of applied water was greater in the former than in the latter (Figure 1). The reference equation would allow estimating the effect of the evaporative demand on the WP and the level of water stress in each growing cycle (Figure 2).

Control values were near the reference equation in the three experiments, although different cultivars showed different values in each cycle. The most important differences between the Control equation and the reference equation were found at the beginning of spring 2022, on the first two dates of the experiment. In addition, Control values were greater than the reference equation on some dates, but mainly from DAT 80 in the spring 2020 experiment. Such differences could indicate an over-irrigation period likely related to problems with the limitations imposed by the COVID-19 pandemic. The pattern of Control data were very similar to the ones in the reference equation in the three experiments, although the autumn cycle used a bigger fruit cultivar than the other two experiments. The comparison with the reference equation suggested that the experiment in spring 2020 presented less severe water stress conditions than the one in spring 2022 in Deficit plants. In both seasons, values slightly lower than −0.8 Mpa were measured in Deficit plants, but they were less persistent in 2020 than in 2022. Absolute minimum values in the Deficit treatment were more negative in 2022 (−0.93 Mpa) than in 2020 (−0.84 Mpa), although both were applied to the same kind of cultivar (cherry type). During spring 2020, Deficit plants presented significantly higher negative values than Control plants at the beginning and end of the experiment. In spring 2022, although the number of days with significant differences was fewer than in 2020, Deficit plants tended to have more negative WP values than Control plants from 50 DAT until the end of the experiment. The pattern of WP in autumn 2022, the one with the biggest fruit size cultivar, was very different from the other two experiments, with a continuous recovery in both treatments from 50 DAT. Only at the beginning of this autumn cycle was WP more negative than −0.8 Mpa in Deficit plants, similar to the ones measured in the two experiments using the cherry type. The number of days with significant differences between Control and Deficit treatments was the greatest in this autumn cycle, although the length of the experiment was the shortest. In all experiments, important differences in WP between treatments were found during the growth in plant height and truss development. All the negative values of WP in the three experiments were measured during the development of the first truss (left arrow, Figure 2).

The estimation of the Stress Integral (SI) showed different water stress patterns in the three experiments (Figure 3).

The lowest SI values were reached in spring 2020, while the maximum values were obtained in spring 2022, both in cherry cultivars. During spring 2020, the SI was significantly higher in Deficit than in Control from the beginning of the experiment. However, the maximum value of SI was around 5 Mpa*day and was obtained just at the beginning. This assumed that the water status, using the suggested reference, was almost equal between Control and Deficit plants most of the time. Conversely, the pattern of SI in spring 2022 increased from 50 DAT and reached maximum values by the end of the experiments (close to 20 Mpa*day). In this cycle, spring 2022, significant differences between Control and Deficit were found at 50 DAT. The SI also increased in Control from the middle of the experiment, but the maximum value was around 5 Mpa*day, clearly lower than in Deficit plants. The pattern in the autumn cycle, with the biggest fruit size cultivar, was somewhere between the ones in previous spring experiments. The SI also increased in Deficit plants from 50 DAT, but it was steady by the end of the experiment (from 80 DAT). The maximum SI in autumn 2022 was around 10 Mpa*day in Deficit plants, double that in spring 2020, but half of that in spring 2022. Significant differences were found between Control and Deficit treatments from DAT 50 in autumn 2022. The development of the first truss occurred when the SI was at its minimum in all experiments and was only considerably different in spring 2020. On the contrary, in spring 2022 and autumn 2022, the beginning of the development of the rest of the trusses occurred under conditions of increasing SI.

Midday net photosynthesis (Pn) did not present major differences on most dates in spring, for the cherry cultivar, and autumn 2022, for the biggest fruit size cultivar; no data were available for spring 2020 (Figure 4). There was only one important difference by the end of the autumn 2022 cycle, likely related to leaf senescence. The seasonal pattern was different for each cycle. In spring, when the cherry cultivar was grown, most values were around 20 μmol m^−2^s^−1^, and only by the end of the experiment had values sharply decreased in both treatments (Figure 4). The decrease around 40 DAT was due to a low radiation day. No important variations were found in these cycles, and only on 60 DAT did Deficit tend to have lower values than Control in a couple of data points. The rest of the data presented almost the same value for both treatments. In the autumn cycle, when the biggest fruit size cultivar was grown, the pattern shows a decrease in Pn values from the beginning of the experiment (Figure 4). These initial measurements were slightly greater than in the Spring cycles, but they presented a progressive decrease in both treatments from 40 DAT, sharper on the last two dates. Only on the last date were Control data significantly higher than Deficit data, although values were very similar on almost all dates.

Differences in plant height between treatments were low in the three experiments (Figure 5). Plants in both treatments reached their maximum height at the same time in the three cycles. In both spring cycles, when cherry cultivars were grown, maximum height was achieved at around 90 DAT.

But only significant differences were found in spring 2020, with greater values in Deficit than in Control plants on a few dates. The autumn cycle, when the biggest fruit-size cultivar was grown, presented a slightly different pattern in comparison with the spring ones. Deficit plants tended to have a slightly lower height than Control, with several dates showing important differences. In this cycle, height in both treatments increased quickly at the beginning of the experiment until 50 DAT, when the growth rate strongly decreased and maximum height was a bit shorter than in the spring cycle.

The plant development indicator estimated not only the plant vegetative growth and integrated height but also the leaf area and number of leaves. The pattern of this indicator was affected by the irrigation treatments (Figure 6).

The decrease in this measurement over the years was due to the removal of shriveled leaves at the base of the plant. Maximum values of plant development were obtained in spring 2020 because the COVID-19 pandemic limited the number of days when leaves could be removed and, also, the number of measurements. However, in this first experiment, Deficit plants tended to have lower values of plant development from the beginning, with significant differences around 50 DAT. On the other hand, no important changes between treatments were found in spring 2022, with almost the same percentage being maintained until 90 DAT. From that date on, Control plants tended to have a slightly greater percentage than Deficit ones, but with very low, non-significant differences. In both spring cycles, cherry cultivars were grown, but this indicator showed the greatest differences. The greatest differences within a season were measured in autumn 2022, when the biggest fruit-size cultivar was grown. Control plants showed values considerably greater than Deficit from 40 DAT, and only the removal of leaves slightly reduced such differences. In this cycle, the pattern was similar to the height one (Figure 5, autumn 2022), with a great increase until 50 DAT, but the rate became slower from that date onwards until the end of the experiment and almost null in Deficit plants.

### 2.2. Yield Response

The pattern of flower and fruit appearance in two different trusses in the spring 2022 cycle was slightly affected by irrigation treatments (Figure 7). The development of the first truss occurred between 23 DAT (see Figure 4, central) and 80 DAT (Figure 7a). The number of fruits increased over time in both treatments, reaching a maximum of 14 fruits per truss. This is the typical number of fruits in cherry cultivars. The pattern of both treatments in this first truss was almost equal in flowers and fruits, with the same amounts and dates. Slight differences were found in the 5th truss (Figure 7b). The development of this truss was from 58 DAT (see Figure 4, central, red arrow) to 107 DAT (Figure 7b). The number of flowers was almost the same in both treatments on all dates, except at around 95 DAT, when significantly fewer flowers were measured in Deficit than in Control plants. Conversely, the number of fruits was very similar, with no significant differences between treatments. The pattern of developed fruits was also almost equal, with no major delays.

The number of harvested fruits in the three experiments presented a similar pattern in both treatments (Figure 8).

The harvest period lasted 45 days, except in spring 2020, when it was slightly longer (55 days), although the cultivar in autumn 2022 had the biggest fruit size, and the other two seasons were cherry types. The beginning of the harvest period occurred on the same date for both treatments in all experiments. The pattern was different in each experiment but almost equal between treatments. No significant differences were found in the number of harvested fruits. On the other hand, the individual fruit weight was considerably reduced in Deficit plants on some dates in spring 2022 and autumn 2022, but not in spring 2020. In the spring cycles, with cherry cultivars, individual fruit weights were around 15 g fruit^−1^, with small changes throughout the experiments. Only three dates with significantly lower weights were found in spring 2022, although in most of the harvest period, Deficit tended to have lower values than Control in both spring cycles. The maximum weight reduction was 15% and 25% of Control in 2020 and 2022, respectively. The average weight reduction throughout the harvest period was 7 and 15% in 2020 and 2022, respectively. In autumn 2022, the cultivar had the biggest fruit size, but significant differences were found only on the last date. However, except for the first sample, Deficit tended to have a lower weight than Control in all of them. The maximum weight reduction was on this last date, with a 30% decrease, and the average for the experiment overall was 14%. Both reduction rates were very similar to the ones obtained with the cherry cultivar in spring 2022.

The cumulative yield was not significantly affected in any of the experiments (Figure 9). Its increase was similar throughout the experiment in both cherry cultivars, although the yield obtained in 2022 was lower than in 2020. In the experiment with the biggest fruit size cultivar, autumn 2022, yield tended to have lower values in Deficit than in Control plants from 90 DAT, although no significant differences were found. The reduction of yield in Deficit plants by the end of both spring cycles was similar: 12% in 2020 and 13% in 2022. In 2020, Deficit plants tended to have lower values on the last 3 harvest dates, while in 2022, this decrease was only observed at the end of the experiment. On the contrary, in autumn 2022, Deficit plants had a final reduction in yield that reached 24%, almost double the ones obtained in cherry cultivars.

The level of soluble solids in fruit is important to increase the flavor of tomatoes. Each experiment presented a different amount of TSS but a similar pattern between treatments (Figure 10).

Cherry cultivars in spring cycles presented the highest values of TSS, greater in 2022 than in 2020. In spring 2020, Deficit fruits presented significantly greater TSS than Control on two dates (157 and 164 DAT), but on the rest of the dates, the values were very similar. On the other hand, in spring 2022, although only two significant differences were found (108 and 128 DAT), Deficit fruits tended to have clearly higher values than Control on most of the dates. The experiment in autumn 2022, the one with the biggest fruit size cultivar, presented the lowest TSS values. Although no important differences were found, Deficit fruits in this experiment tended to have greater values than Control ones.

## 3. Discussion

Water stress conditions in all experiments were moderate because the gas exchange was not significantly reduced (Figure 4), even though the water potential was affected (Figure 2). Tomato is considered a non-isohydric species, which means that, under water stress conditions, leaf water potential gets affected earlier than gas exchange, which also suffers a delay in recovery [22]. Although severe water stress was not imposed, moderate water stress conditions could affect different processes related to yield. Vegetative growth was considerably impacted, and this was more evident in development than in height (Figure 5 and Figure 6). Hsiao [4] reported growth as the most sensitive physiological response to drought conditions and very strongly related to yield response. However, the level, duration, and time point of this water stress could reduce the final effect on yield response. In this way, applied water would not be sensitive enough to manage deficit irrigation. Current data on applied water (Figure 1) suggested a more severe water stress than in fact was measured, according to yield and water relations results (Figure 2, Figure 3, Figure 4 and Figure 9). Therefore, using indicators to evaluate water stress levels was better than a strategy based on applied water. The lack of agreement between irrigation works based on applied water supports this conclusion [6,9,11,12,13]. Moreover, recently, Liu et al. [30] concluded that a combination of frequency and deficit amount of irrigation (the most frequent with 70% Etc) would be the best recommendation for tomatoes. This latter work is actually a selection of the best water stress management strategies. A similar conclusion would be reached using the meta-analysis, which concluded that soil texture was one of the most important factors in the results of tomato irrigation [14]. In the current work, the indicator that better described the pattern of water stress was the SI (Figure 3), because it includes the impact of the duration. Minimum values of WP were not coincident with differences in truss development, fruit size, TSS, and yield (Figure 7, Figure 8, Figure 9 and Figure 10). There are several examples in the literature about the limitations of absolute values of WP as indicators. More negative values (around −1.2 Mpa) than in the current work caused a sharp decrease in yield when they were applied throughout the season [23], but not if they were imposed before the first truss development [15]. Leaf water potential around −1 Mpa for short periods in the middle [18] or at the end of the cycle [7] did not reduce yield either. The current work did not allow selecting a threshold value in SI that could be useful for deficit irrigation scheduling because of limited data. But the improvement of the approach in the SI estimation with the use of a reference equation would increase the accuracy of the water stress management. The experiments with lower SI (spring 2020; less than 5 Mpa*day) presented lower yield reduction and lower TSS than others, even with the same type of cultivar (spring 2022; more than 15 Mpa*day). Coyago-Cruz et al. [24] reported a greater reduction in cherry tomatoes with higher values of SI than the ones reported in the current work.

The yield response to water stress is a complex issue because it depends on different components. The appearance of the truss and the number of fruits were the most resistant parameters to the level of water stress imposed in the current work (Figure 7 and Figure 8), while fruit size and TSS were the most sensitive (Figure 8 and Figure 10). These results were similar for both types of cultivars, although further work is required to clarify the cultivar effect. In most articles about irrigation experiments, the number of fruits was typically less affected by deficit irrigation strategies than the fruit weight, even under more severe water stress conditions and different cultivars than the ones used in the current work [11,13,15,21,31]. The fruit water transport through the xylem has been recently reported to be greater than expected in tomatoes (around 75% via xylem [32]). Johnson et al. concluded that variations in the stem water potential of tomatoes had an immediate effect on the daily pattern of fruit diameter [33]. This work would explain the changes in the fruit size between dates (Figure 8), because it would be more closely related to the WP than to the SI. Changes in TSS could be affected because of an earlier ripening of the fruit and the cumulative effect of water stress. In this way, the reduction in the number of flowers but not in fruits in the 5th truss of the experiment during spring 2022 (Figure 7) suggested a shorter period of fruit development due to water stress conditions, also reported in other works [21]. The increase in TSS values in the current work occurred in the experiments with a greater SI (Figure 3 and Figure 10). Some studies reported changes in the response to water stress between trusses of the same plant, and such differences were also variable between seasons [13]. This latter work described a sustained deficit irrigation strategy, where these results would suggest a cumulative effect of water stress [13]. Other works suggested that the increase in TSS occurred at the pink stage of fruit ripening and varied between locations, although the same final cutoff period or a moderate irrigation reduction was applied [17]. This latter result also supported the influence of water stress rather than water applied, and it is indicative of the effect of cumulative stress.

The management of water stress based on applied water strategies limits the resilience of growers to severe water scarcity periods. Important reductions in water availability will result in the decision to stop cultivating tomatoes because yields will be strongly reduced. Strong reductions in irrigation will affect tomato yield, but accurate management of water stress could reduce such a response. The current work on different cultivars showed that limited irrigation was associated with a low (not significant) reduction in yield. This deficit irrigation scheduling is also more difficult in greenhouses, where an indeterminate growth pattern is commonly used. Moderate water stress conditions before the first truss and during truss development did not significantly reduce yield but increased TSS (Figure 3, Figure 9 and Figure 10). Although the effect of the cultivar could not be assessed with the current data, the results in terms of percentage yield reduction were similar. Zegbe et al. [15] reported the greatest drought sensitivity of flowering and fruit development in tomatoes. But other authors reported that there was no impact on yield with irrigation reduction during this period [14,18], likely related to moderate or shorter water stress conditions. Therefore, deficit irrigation scheduling based on plant water status would increase the possibility of using limited water resources. In this way, the reference equation and the SI approach presented in the current work would be useful for water stress management. Further work will be required to present thresholds and approaches to managing water stress conditions.

## 4. Materials and Methods

### 4.1. Site and Treatment Description

Three experiments were carried out in a non-heated plastic-covered greenhouse at Escuela Técnica Superior de Ingeniería Agronómica (ETSIA) in Seville, Spain (37°21′ N, 5°56′ W, 33 m. a.s.l.). The radiation transmissibility through the experimental greenhouse was estimated at around 75% of outdoor radiation. Passive ventilation was provided by means of lateral and zenithal windows. The soil was clay loam (21.5% gross sand; 4.5 fine sand; 42.3 limo; and 31.8% clay) with pH 8.11 (measured in water) and 2.5% organic matter. These experiments were conducted during two spring cycles (from March to June) in 2020 and 2022 and one autumn cycle (from September to December) in 2022. In each experiment, tomato cultivars were different, but all of them presented indeterminate growth. The cultivars used in the spring cycles were “cherry type” (cv. Summerbrix in 2020 and Grandbrix in 2022), while a “chocolate Marmande type” was planted during the autumn cycle (cv. Marejada). Plants were grown in a nursery seedling for 30 days and transplanted to the experimental site (2 pl m^−2^) when three or four leaves were developing. They were kept at a height of 2 m and pruned to one axis, with the elimination of secondary stems and basal leaves when they were senescent. The irrigation system consisted of single drip lines 1 m apart from each other, with pressure-compensating emitters (4 L h^−1^) every 0.5 m.

The temperature inside the greenhouse was monitored with a temperature and relative humidity sensor (Atmos-14, Meter, Washington, DC, USA) attached to a datalogger (CR1000, Campbell Sci, Shepshed, UK). Figure 11 shows the seasonal pattern of maximum and minimum temperatures during the three experiments. In the spring cycles, maximum temperatures were around 30 °C until 50–60 days after transplant (DAT), when they tended to increase slightly. Maximum values were recorded at the end of the experiments when the temperature reached more than 40 °C. In both experiments, minimum temperatures presented a similar pattern. At the beginning, they were slightly below 10 °C but then increased to 70 DAT. By the end of the experiment, the values recorded were close to 20 °C. The autumn cycle presented the opposite seasonal pattern. Both maximum and minimum temperatures decreased from 35 and 20 °C, respectively, to steady values at around 70 DAT. On this date, maximum temperatures were around 22 °C until almost the end of the experiment. Minimum temperatures presented greater variations between days, with values around 10 °C. But they decreased clearly from 85 DAT and reached minimum values near 0 °C in the last few days.

The greenhouse was oriented close to a N-S direction, and a radiation gradient was observed from the south side of the experiment to the north part. To reduce the potential impact of these variations in radiation, a block design was selected that would minimize such differences. The experimental design used randomized blocks with two irrigation treatments and four repetitions. Each plot consisted of three lines of plants, with the central one used as a measuring line and the other two as guards. The treatments were:Control. full-irrigated conditions. Applied water was 100% of crop evapotranspiration (ETc), which was estimated with an approach using the crop coefficient (Kc) and potential evapotranspiration (ETo) data [35]. ETo inside the greenhouse was estimated using a radiation approach [36]. This calculation required outdoor radiation data and radiation transmissivity (estimated at around 75%). Outdoor radiation data were obtained from the Andalusian network of agroclimatic data at La Rinconada Station [34]. The daily average was 18.12 MJ m^−2^ in spring 2020, 19.01 MJ m^−2^ in spring 2022, and 10.6 MJ m^−2^ in autumn 2022. Kc data were obtained by Battilani et al. [2].Deficit irrigation. The irrigation management of this treatment was based on leaf water potential measurements. Plants were irrigated as a Control at the beginning of the experiment to ensure an adequate setting. The irrigation treatment started at 35 DAT in spring 2020, 32 DAT in spring 2022, and 16 DAT in autumn 2022. After those time points, irrigation was provided only when average leaf water potential values were between −0.7 and −0.8 MPa. Previous works and data from the literature suggested that such values could be an adequate threshold [7,15,24].

Irrigation scheduling was applied weekly after water potential measurements were taken. The estimation of crop evapotranspiration was obtained, and the applied water was calculated. Irrigation was controlled using a remote programming device (Ciclon. C-146 v. 3.53, Maher, Almeria, Spain).

### 4.2. Description of Measurements

The plant-water relations were characterized using leaf water potential and net photosynthesis. Leaf water potential was measured at midday for one plant per plot. Because tomato leaves are very big, a foliole in the external part of the plant was used for this determination. This indicator was measured using a pressure chamber (model 1000, PMS Instrument Company, Albany, NY, USA). The foliole was cut and introduced in a plastic bag with a small piece of moist paper to transfer to the pressure chamber, which was out of the greenhouse, just by the door. Pressure inside the chamber was increased at a slow rate, according to the recommendations of Turner et al. [37], using dry nitrogen. This gas was selected following Hsiao recommendations [4] because it did not affect the stomata aperture and did not significantly change the humidity inside the chamber. Net photosynthesis was also measured at midday in one plant per plot in a fully expanded, healthy foliole using an infrared gas analyzer (IRGA, CI-340, CID-BioScience, Washington, DC, USA). This IRGA was a portable device that measured CO_2_ with an infrared sensor and H_2_O with a humidity sensor capacitor. The device was calibrated every day for zero values of CO_2_ and H_2_O. The photosynthesis measuring process was a comparison between cycles of measurements of the air composition (amounts of CO_2_ and H_2_O). The air outside of the chamber flowed at a constant rate in and out of the chamber. Photosynthesis values were obtained automatically according to the default equations of the device [38]. The leaf chamber had a surface area of 10 cm^2^, and it was completely covered by the foliole. The device provided PAR data with an external photodiode and temperature data from the chamber (thermocouple) and the leaf (infrared sensor). The COVID-19 pandemic limited the possibility of staff involvement in the implementation of the experiments. Consequently, photosynthesis measurements were not taken in the 2020 season.

The stress Integral (SI) was estimated using single leaf water potential data, according to Myers [26]. This approach was similar to the ones suggested by Rudich et al. [25] and estimated the cumulative effect of water stress. The SI estimated the surface of stress obtained as a sum of the duration of all water potential data. The estimation assumed that water potential values between two consecutive measurement days were the average of those two data points. The stress surface will be the sum of all products of water potential data and the number of days between these paired measurements (Equation (1)). Instead of using a water potential level of 0 Mpa as a reference, the water stress surface was referenced to a full irrigated value (“r”, Equation (1)). The approach of Myers [26] suggested the most positive water potential of the experimental data as a reference value. However, this value would not ensure full irrigated conditions, and it would change on each date based on the evaporative demand. For other species, reference equations have been estimated with full irrigated water potential values from several experiments [39,40]. These reference equations were the relationship between water potential and one climatic data, such as temperature [39] or vapor pressure deficit (VPD [40]). This approach to estimating the effect of evaporative demand is very common in this and other water status indicators, such as maximum daily shrinkage [27,29]. In the current work, this reference equation was obtained from previous irrigation experiments on tomatoes (WP = −0.018×max; unpublished data). Then, the SI was estimated with a derivate equation from Myers [26], which included a variable reference value:SI = |∑ (WP − r) ∗ n|(1)
where:WP was the average measured leaf water potential between two consecutive sample dates (MPa).R was the average reference value obtained from a reference equation on these two dates (WP = −0.018*Tmax, where Tmax is the daily maximum temperature on each date).N was the number of days between the sample dates.

Vegetative growth was characterized by plant height and plant development. Plant height was measured weekly for three plants per plot. Plant development was estimated weekly as the percentage of plants that covered a vertical 2 m^2^ square. A white polystyrene square of 1 m^2^ was placed behind the measured row, and a picture was taken with a smartphone. This picture was analyzed using the Canopeo app (free software from Oklahoma State University), which estimated the surface area covered by the plants.

Truss development was measured only in the spring 2022 experiment. In one randomly selected plant per plot, the first and fifth trusses were marked, and the number of flowers and fruits was counted. These measurements were carried out throughout the truss development until the number of flowers was zero and harvest started. In all seasons, the yield of each plot was weighed weekly only for plants in the central row. Only fruits that were considered to be in an adequate ripening stage for consumption were harvested. The average number of fruits per plant and the weight of individual fruits were estimated from the total number of each plot and the number of harvested plants. A sub-sample of three fruits per plot was used to measure total soluble solids (TSS) using a hand-refractometer RHC-200ATC (Huake, China).

All statistical analyses were calculated using the Statistix program (SX 8.0, Analitycal Software, Tallahassee, FL, USA). Data analyses were performed with ANOVA, analyzing the influence of blocks and irrigation treatments. When the influence of irrigation treatment was significant (*p* < 0.05), means were separated with Tukey’s test using the same *p*-level (*p* < 0.05). Data normality was verified with the Shapiro–Wilks test and homoscedasticity with the Bartlett test. Data independence was assumed by the experimental design and data collection. The number of samples measured is specified in the text and figures.

## 5. Conclusions

Moderate water stress throughout the season did not significantly reduce yield in any of the cultivars used. But some yield components were affected, which would explain the trend toward lower yields in Deficit plants. Fruit size was the most sensitive to water stress, and the reduction was likely affected by the WP at a given time instead of cumulative water stress. Fruit number and truss development were hardly affected by the water stress level. These results suggest that deficit irrigation scheduling for indeterminate-growth tomatoes would be possible with accurate management of the water stress. The level of water stress also increased TSS in the fruit, but only in the plants, which presented the greatest SI values. The current SI approach could be useful for deficit irrigation scheduling. However, further work will be required to determine thresholds and differences between cultivars.

## Figures and Tables

**Figure 1 plants-13-00128-f001:**
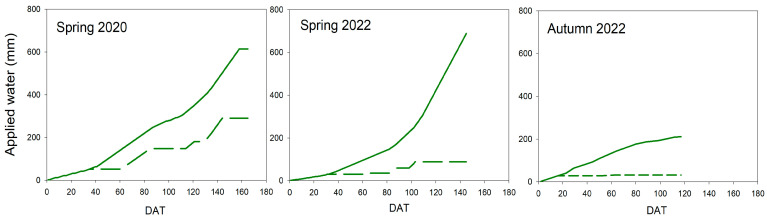
Pattern of applied water throughout the experiments. Solid line, Control; Dash line, Deficit.

**Figure 2 plants-13-00128-f002:**
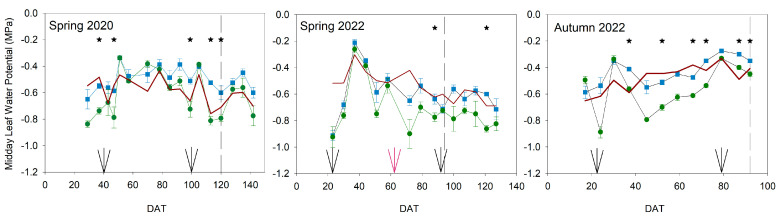
Pattern of midday leaf water potential throughout the experiments. Blue square: Control; Green circle: Deficit irrigation. Each symbol is the average of 4 data. Vertical bars represent the standard error. The solid red line represents data from the reference equation. Dash lines show the date of maximum height. Black arrows mark the beginning of the development of the 1st and last truss. Red arrows indicate the development of the 5th truss in Spring 2022. Asterisks indicate significant differences on that date (*p* < 0.05, ANOVA).

**Figure 3 plants-13-00128-f003:**
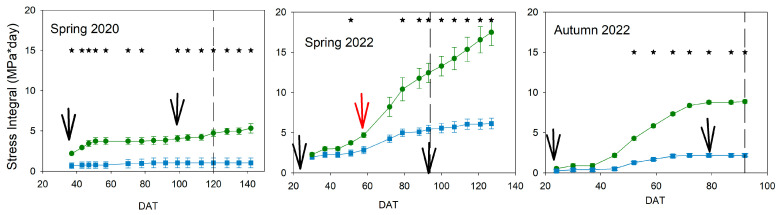
Pattern of the stress integral throughout the experiments. Blue square, Control; Green circle, Deficit irrigation. Each symbol is the average of 4 data. Vertical bars represent the standard error. Dash lines show the date of maximum height. Black arrows mark the beginning of the development of the 1st and last truss. Red arrows indicate the development of the 5th truss in Spring 2022. Asterisks indicate significant differences on that date (*p* < 0.05, ANOVA).

**Figure 4 plants-13-00128-f004:**
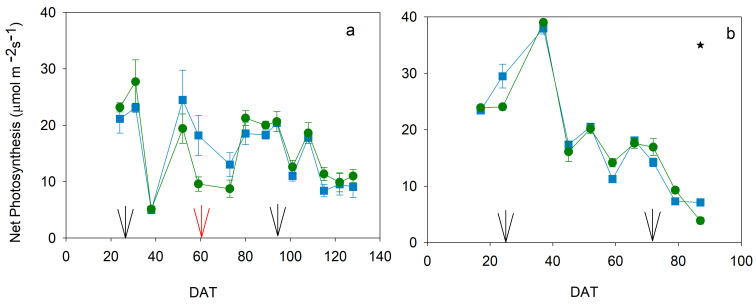
Pattern of midday net photosynthesis during the spring (**a**) and autumn (**b**) cycles in 2022. Blue square: Control; Green circle: Deficit irrigation. Each symbol is the average of 4 data. Vertical bars represent the standard error. Black arrows mark the beginning of the development of the 1st and last truss. Red arrows indicate the development of the 5th truss in spring 2022. Asterisks indicate significant differences on that date (*p* < 0.05, ANOVA).

**Figure 5 plants-13-00128-f005:**
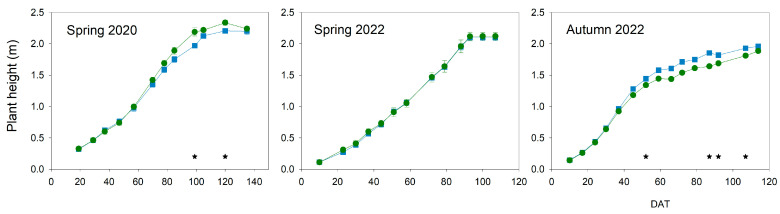
Pattern of plant height throughout the experiments. Blue square: Control; Green circle: Deficit irrigation. Each symbol is the average of 4 data. Vertical bars represent the standard error. The symbol was greater than the vertical bar on the dates when they are not presented. Asterisks indicate significant differences on that date (*p* < 0.05; ANOVA).

**Figure 6 plants-13-00128-f006:**
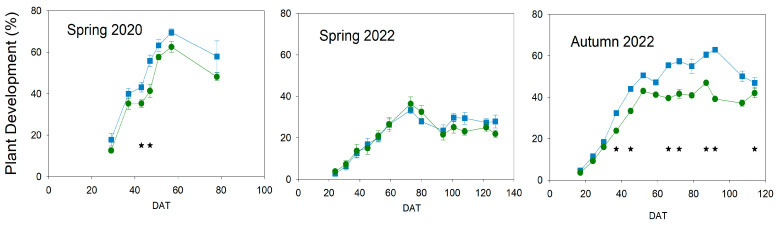
Pattern of plant development (%) throughout the experiments. Blue square: Control; Green circle Deficit irrigation. Each symbol is the average of 4 data. Vertical bars represent the standard error. Asterisks indicate significant differences on that date (*p* < 0.05, ANOVA).

**Figure 7 plants-13-00128-f007:**
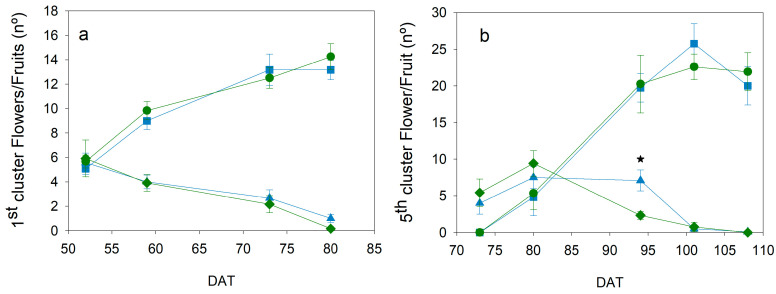
Pattern of fruit and flower development in (**a**) the first truss and (**b**) the fifth truss during the experiment of the spring cycle 2022. Blue symbols are Control (squares, fruits, triangles, and flowers); Green symbols are Deficit irrigation (circles, fruits, diamonds, and flowers). Each symbol is the average of 4 data. Vertical bars represent the standard error. Asterisks indicate significant differences in the number of flowers on the date where they are presented (*p* < 0.05, ANOVA). No significant differences were found in the number of fruits.

**Figure 8 plants-13-00128-f008:**
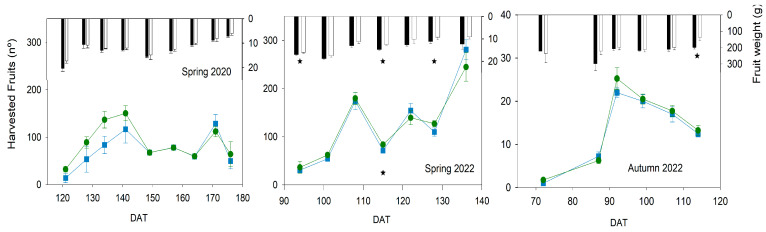
Pattern of harvested fruits (points) and fruit weight (bars) throughout the experiments. Blue square and black bars, Control: Green circle and gray bars, Deficit irrigation. Each symbol is the average of 4 data. Vertical lines represent the standard error. Asterisks indicate significant differences in harvested fruits (bottom) and fruit weight (top) on that date (*p* < 0.05, ANOVA).

**Figure 9 plants-13-00128-f009:**
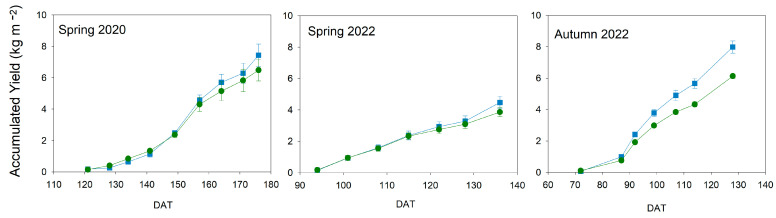
Pattern of cumulative yield throughout the experiments. Blue square: Control; Green circle: Deficit irrigation. Each symbol is the average of 4 data. Vertical bars represent the standard error. No significant differences were found.

**Figure 10 plants-13-00128-f010:**
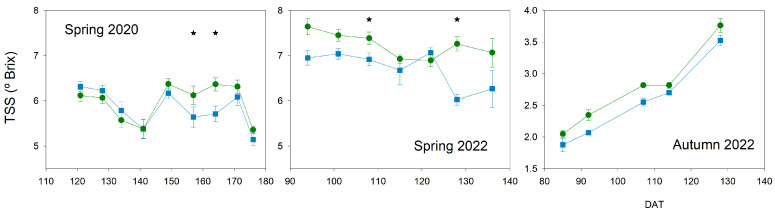
Pattern of total soluble solids (TSS) in fruits throughout the experiments. Blue square, Control; Green circle, Deficit. Each symbol is the average of 4 data. Vertical bars represent the standard error. Stars indicate the date when significant differences were found (*p* < 0.05, ANOVA).

**Figure 11 plants-13-00128-f011:**
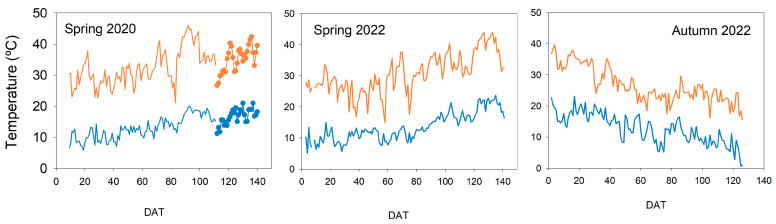
Pattern of maximum and minimum temperature throughout the experiments. Dot data for the spring 2020 cycle were obtained from an external weather station (La Rinconada station, [34]).

## Data Availability

We will include a link to the data, available in the University of Seville repository.

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
