# Peer review of "Moderate Water Stress Impact on Yield Components of Greenhouse Tomatoes in Relation to Plant Water Status"

_plants, 2024, doi:10.3390/plants13010128_

Round 1
Reviewer 1 Report
Comments and Suggestions for Authors
The submitted manuscript to PLANTS-MDPI entitled “Control crop water stress as tool to optimize the quality of new tomato varieties” is interesting to investigate. BUT, following are the comments that need to be addressed:
Why do the authors only choose one drought level?
Line 19: Incomplete sentence.
The data recorded is not ENOUGH!!! At least, there should be some stress biomarkers and antioxidants.
The objectives are not clear, and the research gap was not mentioned either in the introduction section.
Author Response
Thank you very much for your review, we think our manuscript has improved thanks to your comments.
We have improved the introduction with new references.
We have changed the conclusions.
Why do the authors only choose one drought level?
Because the space in the greenhouse was limited. We prefer to increase the number of plant and repetition and use only one drought level.
Line 19: Incomplete sentence.
Changed
The data recorded is not ENOUGH!!! At least, there should be some stress biomarkers and antioxidants.
Of course data about biomarkers and antioxidant will be very interesting however the capacity of obtain data is limited. The article is focused on the irrigation scheduling. In this way, data of biomarkets and antioxidant, though interesting, are not indispensable for water management.
The objectives are not clear, and the research gap was not mentioned either in the introduction section.
We have changes the objective and we think that research gap is, now, enough clear.
Reviewer 2 Report
Comments and Suggestions for Authors
The aim of this work was to evaluate thetomato yield response to deficit irrigation scheduling using the leaf water potential and stress integral during three seasons. Two Spring and one Autumn cycles were evaluated with different fruit size cultivars. However, in the previous works (doi: 10.1016/j.agwat.2018.10.020 - Agricultural Water Management and doi: 10.3390/agronomy13020563 - Agronomy) the authors showed the results of a very similar experiments.
What's new here - an idea to use the prediction of the effects of limited watering to maintain tomato yield in greenhouse cultivation at an assumed level of water stress (measured by the water potential of the leaf).
Comparing the impact of full and limited watering of plants on the yield is not enough for the results to be published here. Perhaps they would find interest in a magazine focused on practical aspects of optimizing tomato cultivation under greenhouse conditions.
Author Response
Thank you very much for your review, we think our manuscript has improved thanks to your comments.
The aim of this work was to evaluate the tomato yield response to deficit irrigation scheduling using the leaf water potential and stress integral during three seasons. Two Spring and one Autumn cycles were evaluated with different fruit size cultivars. However, in the previous works (doi: 10.1016/j.agwat.2018.10.020 - Agricultural Water Management and doi: 10.3390/agronomy13020563 - Agronomy) the authors showed the results of a very similar experiments. What's new here - an idea to use the prediction of the effects of limited watering to maintain tomato yield in greenhouse cultivation at an assumed level of water stress (measured by the water potential of the leaf).
The current work try to validate and improve this previous work. The management of water stress based on water status measurements is very difficult and we have tried to check if it would possible the use of the mange suggest for this paper. In addition, we have include a change in the estimation of SI which is the most important novelty in comparison with the previous work. This change (the use of a reference equation) is a very new approach. In addition, the evaluation of different cultivars also is very interesting because would permit to check if this management could be use in quite different fruit size cultivars.
Comparing the impact of full and limited watering of plants on the yield is not enough for the results to be published here. Perhaps they would find interest in a magazine focused on practical aspects of optimizing tomato cultivation under greenhouse conditions.
We disagree. This article is not just a comparison between two treatments of irrigation form the point of view of yield. We proposed a framework, not new but very uncommon for irrigation scheduling. We showed yield and yield components in relation to the pattern of water stress. Although with a limited number of treatments, data are very interesting from our point of view because present different cultivars (in fruit size) and different cycles. Probably, we can not separate the effect of cultivar and irrigation but we show that, in fact, the variations between cultivars were very small. In our opinion this is one of the best journal where this work could be published.
Reviewer 3 Report
Comments and Suggestions for Authors
In lines 15-16 the authors write «The aim of this work was to compare deficit and full irrigated conditions in Spring and Autumn cycles ». « Compare » is not (should not be) the purpose of research. This is a METHOD to achieve a goal. The goal may be, for example, "finding the optimal irrigation technology." I don't understand the term "Treatments were: Control (full irrigated) and Deficit. Deficit was irrigated as the Control ones…” (lines 18-19). What was the control in the experiments? Please expand on the concept: "expected values under full irrigated conditions" (line 22). I cannot understand the logical arrangement of the article in which the authors first present the results (from line 100) and then describe the research methodology (from line 357). In lines 186-187 the authors write: "..because the COVID-19 pandemic limited the number of days… » Very interesting statement…. (sic!!!) It would be more correct to write » .. the Covid-19 epidemic limited the possibility of staff participation in the implementation of experiments and influenced the implementation of the experiment...". Why do the authors present results regarding plant irrigation in the "Materials and methods" section? Why does the publication not describe the type of irrigation system, irrigation technology and irrigation control method (very important)? The authors write about two different species of tomatoes tested, why is this difference not distinguished in the presentation of the results? The article is intended to be of an implementation nature and should contain specific tips for producers in its conclusions. The publication has a very unorganized layout that definitely needs tidying up.
Author Response
Thank you very much for your review, we think our manuscript has improved thanks to your comments.
We have improved the introduction and extended the explanation about the methods that we use in the work. Discussion and results are be rewritten (completely in the case of the discussion). The conclusions have also changed.
In lines 15-16 the authors write «The aim of this work was to compare deficit and full irrigated conditions in Spring and Autumn cycles ». « Compare » is not (should not be) the purpose of research. This is a METHOD to achieve a goal. The goal may be, for example, "finding the optimal irrigation technology." I don't understand the term "Treatments were: Control (full irrigated) and Deficit. Deficit was irrigated as the Control ones…” (lines 18-19). What was the control in the experiments? Please expand on the concept: "expected values under full irrigated conditions" (line 22).
These are part of the abstract. We have changed this, but, please, note that the space of this part of the manuscript is very short (200 word).
I cannot understand the logical arrangement of the article in which the authors first present the results (from line 100) and then describe the research methodology (from line 357).
The publication has a very unorganized layout that definitely needs tidying up.
Material and methods are at the end of the article because this is the organization that the Journal include in the « information for authors ». We apologise for the mistakes and the confunsion in the last version of the manuscript. We have reorganised the information in a clear way (we hope so) and try to be more precise in the explanations
In lines 186-187 the authors write: "..because the COVID-19 pandemic limited the number of days… » Very interesting statement…. (sic!!!) It would be more correct to write » .. the Covid-19 epidemic limited the possibility of staff participation in the implementation of experiments and influenced the implementation of the experiment...".
Changed
Why do the authors present results regarding plant irrigation in the "Materials and methods" section?
We have changed the applied water figure to results.
Why does the publication not describe the type of irrigation system, irrigation technology and irrigation control method (very important)?
In our opinion this is not the most important information about the experimet. The article describe the type of irrigation (drip) and the ditribution of lines and emitters. We also describe the managemnt of water when we describe the treatament. We also included in the new version a description of the control method.
The authors write about two different species of tomatoes tested, why is this difference not distinguished in the presentation of the results?
You are rigth. We have included the reference to the cultivars in the resukts and also in the discussion.
The article is intended to be of an implementation nature and should contain specific tips for producers in its conclusions.
In our opinion, this informarion would be better in the discussion. You are right and we have written a specific part about the implementation in a commercial orchard.
Reviewer 4 Report
Comments and Suggestions for Authors
review plants-2778439
Comments to authors
The authors investigated the variation in leaf water potential of tomato cultivars of different fruit sizes with indeterminate growth and the effect of water stress on yield and Brix based on this variation. The title of the article does not reflect the content, and the manuscript contains a number of shortcomings and inaccuracies. In the results, the discussion chapter requires considerable revision. The results of the spring and autumn seasons cannot be compared as the two different varieties are not included in both seasons. The language in English made the results and discussion very difficult to understand and should be improved.
Detailed comments
Results presented for two different growing seasons and varieties do not support the conclusion. In autumn, there is a big difference between the spring weather data and the amount of water applied, and the variety with large weight fruits is not included in the spring experiment.
I propose to evaluate only the cherry tomato cultivar in the spring trials, specifically highlighting the relationship between stress index (SI) and leaf water potential and through this the relationship between SI and yield and to delete the data from the autumn trial.
-The position and numbering of the figures should be changed: the figure showing the temperature (Fig.1) should be moved to the methodology chapter, where it evaluates the data.
-Fig.6 is shown twice, delete it lines 185-186.
-Not found in Fig. S1, but it was evaluated between lines 162-169.
-The numbering of the figures should be changed according to the order of occurrence in the results, i.e., the leaf water potential indicator should be Fig.1, the stress integral (SI) indicator should be Fig 2, the plant height should be Fig3, etc.
-The relationship between the integrated stress index (SI) and yield is not reliably presented in the results chapter, which is referred to in the discussion chapter.
-In lines 206 and 210 there is a reference to Figure 3a and Figure 3b but this is not in the manuscript only Figure 3.
-Lines 245-247: "exept in Autumn 2022 when..." the crop does not grow in deficit plants (see Fig.9). Check and correct the sentence.
-Line 260: "Control on two dates" which date exactly? It should be marked.
-Lines 274-278: Check the sentence, there is no evidence for this statement in the results.
-Lines 303-304: This is a hypothesis, not supported by the results.
-Lines 310-314: This statement is debatable, the yield loss associated with SI is not shown in the results only cumulative yield by DAT (Figure 9) and SI by DAT (Figure 4).
Conclusion:
-Lines 470-471: It is not possible to compare the small and large fruit weight varieties as they were not tested under the same conditions.
-In text, references should be corrected throughout the manuscript: the authors should be followed by the reference number in brackets e.g. Marouelli and Silva (18), Rudich et al. (23) suggested etc.
-The numbers in ha-1, fruit-1, m-2 units should be placed in the superscript throughout the manuscript.
Comments on the Quality of English Language
The English language made the results and discussion very difficult to understand and should be improved.
Author Response
Thank you very much for your review, we think our manuscript has improved thanks to your comments.
We have changed the Introduction and Material and Methods sections. We have included new reference and explain wider the methods used. Results and Discussion have also been re-written (totally the part of Discussion). We have changed the presentation of the data in a way that, in our opinion, is clearer and in agreement with the recommendation of this reviewer. Conclusions are also different according to the changes in the discussion. The manuscript has been editing, again, for the English.
The title of the article does not reflect the content, and the manuscript contains a number of shortcomings and inaccuracies.
We have changed the title and eliminate the shortcomings and mistakes.
In the results, the discussion chapter requires considerable revision. The results of the spring and autumn seasons cannot be compared as the two different varieties are not included in both seasons.
We have changed the results presentation and completely re-written the discussion section. We have completely changed the presentation of the results in the way that this reviewer suggest. However, we do not agree with the suggestion of eliminate the Autumn cycle. In have included in the discussion several sentence in which we indicate that the effect of cultivar cannot be separate. But we think it is very interesting to present data of different conditions (cultivar and cycle) which could complement the data of Spring cycle in cherry cultivars.
The language in English made the results and discussion very difficult to understand and should be improved.
We apologise for all mistakes in English. The manuscript was revised before sending to the journal and it will revise again.
Detailed comments
Results presented for two different growing seasons and varieties do not support the conclusion. In autumn, there is a big difference between the spring weather data and the amount of water applied, and the variety with large weight fruits is not included in the spring experiment.
We have changed the conclusions according to this comment. However, because the irrigation treatments and the comparison are based on water status instead of applied water the limitations that the reviewer presented here are, in our opinion, small. We can accept that the comparison between cultivars is not possible (we have include this idea un the manuscript) but we think that the presentation of data and the comparison of results are quite interesting, mainly for future works.
I propose to evaluate only the cherry tomato cultivar in the spring trials, specifically highlighting the relationship between stress index (SI) and leaf water potential and through this the relationship between SI and yield and to delete the data from the autumn trial.
The reviewer is right, there is not data of big size cultivar in Spring cycle. However, in our opinion, these data are very interesting and support part of the conclusions. The differences between cycle are the expected because of the greater evaporative demand between one and other. However, the current work wants to focus on water stress level more than applied water. Moreover, the differences in fruit size between cultivars would be very interesting in relation to the number and fruit growth. In our opinion, all these data would permit a comparison of difference responds of tomato crop.
-The position and numbering of the figures should be changed: the figure showing the temperature (Fig.1) should be moved to the methodology chapter, where it evaluates the data.
Changed
-Fig.6 is shown twice, delete it lines 185-186.
Delete. Thank you very much.
-Not found in Fig. S1, but it was evaluated between lines 162-169.
We are so sorry. These are the data of gas exchange, we included as supplementary material and probably they were noy upload. We have now decided to include in the manuscript
-The numbering of the figures should be changed according to the order of occurrence in the results, i.e., the leaf water potential indicator should be Fig.1, the stress integral (SI) indicator should be Fig 2, the plant height should be Fig3, etc.
Changed.
-The relationship between the integrated stress index (SI) and yield is not reliably presented in the results chapter, which is referred to in the discussion chapter.
Unfortunately, we have not found a clear relationship. The influence of water stress in yield is not easy because they could affect different processes. From our point of view, the present work support that cumulative rather than punctual water stress conditions affected processes such as truss development, fruit size and yield. We have focus is this interpretation rather than the estimation of threshold values.
-In lines 206 and 210 there is a reference to Figure 3a and Figure 3b but this is not in the manuscript only Figure 3.
This is a mistake, we have solved it. We are sorry.
-Lines 245-247: "exept in Autumn 2022 when..." the crop does not grow in deficit plants (see Fig.9). Check and correct the sentence.
We have corrected the sentence
-Line 260: "Control on two dates" which date exactly? It should be marked.
We have included the date in the manuscript. They are market in the figure with an asterisk.
-Lines 274-278: Check the sentence, there is no evidence for this statement in the results.
Yes, there is evidence. This is a misunderstanding because the Photosynthesis figure was not presented. In Fig 3 we can observed that there is a water stress because WP is more negative in Deficit plants. These results occur in both experiment in 2022. However, there is not any reduction in photosynthesis (Fig. 4), then gas exchange, this is the common non-isohidric respond describe in the literature.
Minimum WP occurred in all experiments but they do not affect yield
-Lines 303-304: This is a hypothesis, not supported by the results.
We have rewritten these sentences.
-Lines 310-314: This statement is debatable, the yield loss associated with SI is not shown in the results only cumulative yield by DAT (Figure 9) and SI by DAT (Figure 4).
We are going to rewritten the sentence. The reviewer is right, it is debatable but in our opinion we can, at least, include in the discussion. We also want to remark that the influence of water stress in yield is variable and it is very difficult to isolate only one effect from other. More when, as the reviewer point out, we used different cultivars. However, from our point of view, the current work presented data that support:
- A short decrease in WP had no effect in truss and fruit development
- The decrease in yield and fruit size and also the changes in TSS were related with water stress. Perhaps also with other things such as temperature in Autumn. Perhaps the respond changed with the cv. But the effect of water stress was demonstrated.
- The most important decrease in yield were related with fruit size and coincident with the experiment with greater SI. It is true that we have no regression that demonstrated this but we have enough data to suggest that SI could be an interesting indicator. This conclusion is not final, not with 100% of certainty but, in our opinion, enough to present as conclusion with “further work will need to check this result”.
Conclusion:
-Lines 470-471: It is not possible to compare the small and large fruit weight varieties as they were not tested under the same conditions.
We have rewritten the conclusions.
-In text, references should be corrected throughout the manuscript: the authors should be followed by the reference number in brackets e.g. Marouelli and Silva (18), Rudich et al. (23) suggested etc.
Corrected
-The numbers in ha-1, fruit-1, m-2 units should be placed in the superscript throughout the manuscript.
Corrected
The English language made the results and discussion very difficult to understand and should be improved.
We apologise for the mistake in English. We have rewritten part of the manuscript and change the no clear expression. The new version has been checked again.
Round 2
Reviewer 2 Report
Comments and Suggestions for Authors
The work has been thoroughly improved, including: by more clearly emphasizing the main purpose of the research and specifying the most important achievements. However, it still contains some shortcomings that require clarification. These are:
1. On the page 8, in Figure 8, the left Y axis in the two boxes (for Spring 2020 and Spring 2022) corresponds to the number of harvested fruits or fruits fresh weight? See on Fig. 7 and clarify the Y values.
2. On the page 9, in line 541 the followed sentence “The level of soluble sugars in fruit is important to increase the flavour of tomatoes. Each experiment presented a different amount of TSS but a similar pattern between treatments (Fig. 10).” And in legend for Figure 10 is “Pattern of total soluble sugars (TSS)…” Moreover, in Abstract, on the page 1, in line 22 the following mention “Fruit size and total soluble sugars” was found. However, in Material and Methods, on the page 14, in lines 1081-1082, the Authors stated that “A sub-sample of three fruit per plot was used to measure total soluble solids (TSS) using a hand-refractometer RHC-200ATC (Huake, China).” Thus, what was measured – total soluble sugars or total soluble solids?
Author Response
Reviewer 2
- On the page 8, in Figure 8, the left Y axis in the two boxes (for Spring 2020 and Spring 2022) corresponds to the number of harvested fruits or fruits fresh weight? See on Fig. 7 and clarify the Y values.
They correspond to the number of fruit harvested. As we indicated in the text they were cherry cultivars and the number odf fruit s is very high. Please note, that the data of the Y axis at Fuig 7 is only ONE truss. Data on Fig 8 are the total amount that were harvested at that date.
- On the page 9, in line 541 the followed sentence “The level of soluble sugars in fruit is important to increase the flavour of tomatoes. Each experiment presented a different amount of TSS but a similar pattern between treatments (Fig. 10).” And in legend for Figure 10 is “Pattern of total soluble sugars (TSS)…” Moreover, in Abstract, on the page 1, in line 22 the following mention “Fruit size and total soluble sugars” was found. However, in Material and Methods, on the page 14, in lines 1081-1082, the Authors stated that “A sub-sample of three fruit per plot was used to measure total soluble solids (TSS) using a hand-refractometer RHC-200ATC (Huake, China).” Thus, what was measured – total soluble sugars or total soluble solids?
You are right the correct is soluble solids. We apologise for the mistake
Reviewer 4 Report
Comments and Suggestions for Authors
revised 2 plants-2778439
Comments to authors
Significant corrections have been made to the manuscript but the references in the text have not yet been corrected, need to be corrected.
-The reference number should be written after the name of the cited author in bracket, not at the end of the sentence e.g. in line 142 Thompson et al (20) proposed…, line 143 Marouelli and Silva (20) changed…., line 190 and 195 Rudich et al (25)
This is the way to correct line 196 and 652, lines 625 and 976, line 633, line 717, line 744.
-Line 131 "small o no" may be a typo? Correct it.
Author Response
-The reference number should be written after the name of the cited author in bracket, not at the end of the sentence e.g. in line 142 Thompson et al (20) proposed…, line 143 Marouelli and Silva (20) changed…., line 190 and 195 Rudich et al (25)
This is the way to correct line 196 and 652, lines 625 and 976, line 633, line 717, line 744.
Correct them, thank you very much
-Line 131 "small o no" may be a typo? Correct it.
You are rihght. It is “or”